# Relationship between Learning and Psychomotor Skills in Early Childhood Education

**DOI:** 10.3390/ijerph192416835

**Published:** 2022-12-15

**Authors:** José Manuel Alonso-Vargas, Eduardo Melguizo-Ibáñez, Pilar Puertas-Molero, Federico Salvador-Pérez, José Luis Ubago-Jiménez

**Affiliations:** 1Faculty of Education Sciences, Department of Didactics of Musical, Plastic and Corporal Expression, University of Granada, 18071 Granada, Spain; 2Universidad Internacional de la Rioja (UNIR), 26006 La Rioja, Spain

**Keywords:** psychomotricity, learning, learning behaviours, active methodologies, early childhood education

## Abstract

Psychomotor skills are, among others, an aspect particularly valuable for structuring the teaching–learning process of infant schoolchildren. For this reason, a study was carried out with the aim of describing and comparing the socio-demographic, psychomotor, and learning levels of schoolchildren in the second stage of infant education. Ninety-five pupils from the second cycle of infant education in the capital of Granada took part in this study. A sociodemographic questionnaire, the movement assessment battery for children-2 (MABC-2), and the preschool learning behaviour scale (PLBS) were used to collect data. The main results show that manual dexterity appears as the main motor factor and similar figures in the three dimensions of learning behaviours. On the other hand, balance and learning behaviours were higher in 6-year-old schoolchildren. In terms of gender, girls obtained higher values for the level of the learning behaviour variables. A positive correlation was found between the dimensions of learning and motor activity.

## 1. Introduction

Early childhood education is the most important stage in a person’s life cycle [1]. This period covers the first five years of life, being the phase of greatest intellectual progression of the human being [2,3].

This stage of change, where the child leaves home and enters a school institution, begins to socialise among peers, and to comply with schedules, rules, or regulations, requires a high cognitive effort as the infant will begin to interpret, analyse, and predict among other complex actions [4]. On the other hand, important for the objective of encouraging the child’s learning is the involvement of the teacher and his or her contribution to the overall development of the student beyond the cognitive aspect [5], as well as for the proper development of the child in the centre [6].

The characteristics of this vital period mean that, from the educational environment, the child’s innate abilities can be strengthened to the maximum through experiences with which the child will develop new skills or affinities. Therefore, the care given to this stage will be decisive, in accordance with Alliaume [7]; every protective measure must safeguard integrity, i.e., provide support in the cognitive, physical, and socio-affective aspects, which implement child development. To this must be added the promotion of an academic offer that favours the potentiality of these aspects [8]. On the other hand, neural connections at this stage are another axis to be promoted, as scientific findings confirm that the brain develops and is not born as it is [9]. This evolution occurs even before birth, with an ambiguous interaction between neural connections and experiences and the environment, being the foundation of postnatal learning and memory [10].

Pedagogically, active methodologies are increasingly accepted in the preschool stage [11], in line with Pestalozzi’s approach [12]. The model established by Froebel, based on the child as an active agent, proposes that children should be discoverers and actors in their learning [13]. Piaget, from a psychological and pedagogical approach, emphasises that learning involves sensory-motor skills [14,15]. Likewise, Wallon highlights the motor dimension as a component of the individual [16]. Decroly gives importance to visual, motor, and auditory play among others [17,18]. Research highlights early stimulation as a factor that favours brain plasticity in children [19]. Montessori, in turn, relies on the non-obstruction of the preschooler’s individuality and expressive freedom [20,21,22]. In this sense, the WHO [23] recommends the implementation of active methodologies that favour and encourage the acquisition of physical activity habits that remain over time [24].

At an early age, motor skills are key to the development of psychological functions, and it is through movement that the most basic forms of relationship and communication emerge [25]. Motor acts such as standing and grasping only occur in humans who have thrived in civilization, so that, in addition to inherited abilities, environmental behaviours and learning are necessary for them to develop [26].

Psychomotor practice is also inherently playful, procedural, and enjoyable, moving away from the monotony and sedentary nature of many school activities, which has a positive effect on children’s stimulation and participation [27]. Well-directed and planned physical education will contribute to physical, social, affective, psychological, and emotional development in the preschool stage, as well as being a good way to generate learning in other dimensions of knowledge [1,28]. Likewise, difficulties or problems in cognitive, academic, socio-emotional, or other skills development may be detected [28].

Recent studies have shown that active learning improves psychomotor skills, teamwork, reflection, participation, autonomy, responsibility, and the acceleration of meaningful learning [29,30]. In other words, an increase in physical activity is directly related to an increase in physical, cognitive, emotional, and social capacity [28,31], also confirming the effectiveness of the integrated teaching of physical education with areas such as mathematics [32] or languages [27].

There is a high percentage of fine motor delay in children aged 3 to 6 years [33]. At this early stage, the body and movement are determining factors in the learning and development of individuals [34]. Other aspects to be highlighted are its preventive property against difficulties and pathologies derived from a sedentary lifestyle or its palliative condition in terms of the consequences of some disabilities, as well as adherence to sport and a healthy lifestyle [35]. It should be emphasised that the legislative instruments guiding this period of infant education are committed to the development of these aspects, which are present in *Organic Law 3/2020*, of 29 December, which modifies *Organic Law 2/2006*, of 3 May, on education (LOMLOE) [36]. This law reveals the value of physical and healthy activity in the learning and training of schoolchildren.

There is a scarcity of studies and analyses relating to this early stage with respect to others at higher ages. It is important to understand the variables contemplated as aspects of vital importance at this stage, especially those involved in delays in fine motor skills in preschool children, as well as the worrying levels of sedentary lifestyle that currently exist, and their possible influence on learning. This study aims to describe the psychomotor and learning behaviour levels in schoolchildren in infant education and, in turn, to relate these parameters to each other.

## 2. Materials and Methods

### 2.1. Design and Participants

This study presents a descriptive and cross-sectional design that analyses a total sample of 95 early childhood education students, from the same school, in the centre of the city of Granada (Spain), aged between 4 and 6 years, from which, 50.5% (N = 48) were boys and 49.5% (N = 47) girls. The characteristics of the geographical area indicate a medium-high socioeconomic level. The participants belong to six classes of this school with a ratio of approximately twenty-two students per class. The motor activities carried out by these students at school present a lack in terms of dedication time, this being an aspect to consider.

### 2.2. Instruments and Variables

An ad hoc class questionnaire was used to collect socio-demographic variables such as year of birth, gender, and academic level of the pupil’s parents. The following instruments were used for the analysis of the remaining variables:Movement assessment battery for children 2 (MABC-2), in its Spanish adaptation by Ruiz and Graupera-Sanz [37], intended to collect data on the psychomotor variable. This battery, in its application for the infant stage, is composed of eight tests that assess three main aspects: manual dexterity, aiming and catching, and balance. In this version, a Cronbach’s alpha of 0.81 was obtained for the complete battery.
-Manual dexterity: three tests; inserting coins into the slot of a small box (six coins for 4- and 5-year-olds and twelve coins for 6-year-olds), stringing beads (six beads for 4- and 5-year-olds and twelve beads for 6-year-olds), and drawing lines on paper with a felt-tip pen.-Aiming and catching: two tests; catching a small beanbag and throwing the sack at a target.-Balance: three tests; tiptoeing, balancing on one leg, and jumping on mats.

Furthermore, in terms of reliability analysis, Cronbach’s alpha obtained a value of α = 0.856.

Preschool learning behaviours Scale (PLBS) [38]. It consists of twenty-nine items, which are intended to measure the behavioural variable of children’s learning in terms of three parameters: their motivation for competence, their attitude towards learning, and their attention or persistence. Each section offers three possible answers, almost always, sometimes, and usually not, with the assigned values being 2, 1, and 0, respectively. Of these 29 items, numbers 1, 4, 11, 20, 25, and 28 are positively phrased, the remaining items being negative and therefore of inverted valence. Finally, the reliability analysis obtained a value of α = 0.855.

### 2.3. Procedure

Firstly, a collaboration agreement was reached between the school and the research team. In order to collect the data using the instruments described above, once they had been confirmed and prior to filling them in, the preschool teaching staff were informed, being shown the different tests together with a document justifying the study. The school management was also informed, and a letter was sent to the school, with both parties giving their approval for the study to be carried out. Subsequently, we distributed an envelope for each pupil to the different classrooms, which contained an explanatory document and a sheet containing an explanation of the procedure, the sociodemographic test to be filled in by the pupil’s parent or guardian, the MABC-2 battery of activities, and the PLBS test, which, as stated at the end of the page, would be filled in by the research team. Once the families had returned the document with their part completed, we proceeded to contact the tutors of the preschool groups to find a suitable time to carry out the battery of activities with their pupils and, secondly, to ask for their collaboration in the PLBS test. This research was assessed by the ethics committee from the University of Granada (1478/CEIH/2021). The research team was made up of the authors of this manuscript, who had the support and cooperation of teachers from the educational centre where the study was carried out. The entire process strictly followed the principles of the Declaration of Helsinki.

### 2.4. Data Analysis

The data were processed with the SPSS 25.0 statistical software (SPSS, IBM, SPSS Statistics, v.25.0, Chicago, IL, USA). The normality and homogeneity of variance of the variables were tested using the Kolmogorov–Smirnov test. For the descriptive analysis, a study of frequencies and basic descriptive variables was carried out. Subsequently, contingency tables, T-Student, ANOVA, and Pearson’s bivariate correlations were used for the comparative analysis. To establish statistically significant differences, the Pearson Chi-Square test was used. In this case, the significance level was set at *p* < 0.05.

## 3. Results

Regarding Table 1 the MABC-2, the highest mean value (M = 0.95) is in manual dexterity, followed by balance (M = 0.76), and finally aiming and catching (M = 0.65). For the learning variable, the highest mean value is found in attention/persistence (M = 1.73), followed by attitude towards learning (M = 1.70), and motivation for competence (M = 1.67).

Regarding Table 2 the relationship between the MABC-2 and age, no differences were found in the parameters of manual skills and aiming and catching (*p* ≥ 0.050). However, in balance, statistically significant differences were found (*p* = 0.000), due to the fact that students aged 6 years show a better mean (M = 0.96) than those aged 4 years (M = 0.60).

For the relationship between the MABC-2 and the academic level of both parents, no statistically significant differences were found (*p* ≥ 0.050).

In terms of the relationship (Table 3) between learning and gender, statistically significant differences were found in the parameters of motivation for competence (*p* = 0.020), and attention/persistence (*p* = 0.002), due to a higher mean in the female gender, as can be seen in the following table:

Regarding Table 4 the relationship between learning and age, statistically significant differences were found in the parameters of motivation for competence (*p* = 0.001), attitude towards learning (*p* = 0.000), and attention/persistence (*p* = 0.001), generated by a higher mean in 6-year-olds, as shown in the following table:

Regarding the relationship between learning and the father’s and mother’s academic levels, no statistically significant differences were found (*p* ≥ 0.050).

Regarding the correlations (Table 5), the results indicated that manual skills correlate moderately and positively with the dimensions of learning (E, r = 0.347 **; ACT, r = 0.437 **; AP, r = 0.377 **) in such a way that, as one increases, so do the other dimensions. Likewise, moderate and positive correlations are detected in aiming and catching with the dimensions balance (r = 0.435 **), attitude (r = 0.322 **), and attention/persistence (r = 0.309 **). Similarly, balance correlates with attitude towards learning (r = 0.336 **). Motivation for competence correlates strongly and positively with the dimensions (ACT, r = 0.532 **; AP, r = 0.801 **), just as attitude towards learning correlates with attention and persistence (r = 0.731 **).

## 4. Discussion

The academic level of the parents of the students is mostly university level, which is consistent with those reported by Álvarez et al. [39], Li et al. [40], and Merino et al. [41]. These data do not coincide with the majority in the average academic level reported by Määttä et al. [42] and Martisone et al. [43], with the understanding that the area of Granada where the school is located is of a medium-high socio-economic level, so that families who live in the area and enrol their children in this educational institution have a higher value in this aspect, which coincides with what is stated by OECD [44] in terms of the comparison between rural and urban areas.

Regarding MABC-2, the highest mean value is in manual skills, which is similar to Hirata et al. [45] and Van der Veer et al. [46], explaining this by the educational models based on manual work over other aspects, which, being also motor, are not so important.

In learning, the highest mean value is found in attention/persistence, and these data correspond to those reported by Angelo [47], which indicates the importance given to this ability within the educational institution, as it is a determining factor both in the day-to-day life of the classroom and in the acquisition of knowledge by the pupils. The female gender presents a higher mean in the three subscales of the test, which is in agreement with what has been reported by Johnson [48] and Valiente et al. [49], who indicate that girls have a better ability to manage learning strategies as well as stress control. Schoolchildren aged 6 years showed a higher motivation for competence, attitude towards learning, and attention/persistence, which is postulated in contrast to the findings of Sáez et al. [27] and Schaefer et al. [50], who indicated in their studies that learning behaviours remain constant with increasing age. Thus, it is understood that this discrepancy is caused by the different methodologies and learning strategies used by schools and specifically by the teaching teams.

In balance, differences were found because the 6-year-old students showed a better average, as Reina and de Haro [51] pointed out, when they indicated that physical abilities evolve with age; therefore, the 6-year-old schoolchildren analysed showed better balance than those of younger ages. Manual skills and balance correlate moderately and positively with the dimensions of learning, so that as one increases, so do the others. Likewise, motivation for competence correlates strongly and positively with motor dimensions, and attitude towards learning correlates with attention and persistence, which is in line with Franco et al. [52] and Pizani et al. [53], who noted that motivation towards learning and motor generated an increase in both cases, so that, in agreement with Tandon et al. [54], a quality motor environment has a positive impact on schoolchildren’s learning.

## 5. Conclusions

The following findings were obtained from this descriptive, cross-sectional research study:

The students showed, at the motor level, that the highest dimension was manual skills, with no significant differences in terms of gender in any motor variable. In learning, attention/persistence, attitude towards learning, and motivation for competence obtained similar values, with a higher mean in 6-year-old students. Similarly, balance was higher in schoolchildren of this age. In the female gender, higher values were obtained in all three learning subscales. A positive correlation was found between the dimensions of learning and the motor parameters analysed.

Following the results obtained, the need to promote and encourage physical activity in preschoolers through active methodologies is observed. This will allow, according to the results of this study, the levels of learning behaviours from psychomotor work at school to be improved.

A series of limitations have been evidenced based on the characteristics of this research model. Its cross-sectional, descriptive typology has allowed only one measurement to be carried out at a specific point in time and the relationships between the variables to be observed at that time, resulting in the impossibility of establishing cause–effect relationships that some variables apply to others. On the other hand, it should be stressed that the sample in this study focuses on a very specific type of student and a specific geographical area, which prevents the results obtained from becoming generalisable. Finally, it should be noted that several questionnaires intended for the children’s families were not filled in correctly, which reduced the sample of participants to be analysed.

## Figures and Tables

**Table 1 ijerph-19-16835-t001:** Descriptive data of the study.

Gender			Father’s Academic Level		Mother’s Academic Level		
**Male**	50.5% (*n* = 48)		**University Education**	73.7% (*n* = 70)	**University Education**		85.3% (*n* = 81)
**Female**	49.5% (*n* = 47)		**Baccalaureate**	9.5% (*n* = 9)	**Baccalaureate**		4.2% (*n* = 4)
**Age**			**Secondary Education**	5.3% (*n* = 5)	**Secondary Education**		1.1% (*n* = 1)
**4 years old**	32.6% (*n* = 31)		**Primary Education**	1.1% (*n* = 1)	**Primary Education**		0% (*n* = 0)
**5 years old**	34.7% (*n* = 33)		**Vocational Education**	8.4% (*n* = 8)	**Vocational Education**		6.3% (*n* = 6)
**6 years old**	32.6% (*n* = 31)		**Others**	2.1% (*n* = 2)	**Others**		3.2% (*n* = 3)
**Psychomotricity (MABC-2)**				**Learning (PLBS)**			
**Manual skills**		M = 0.95		**Motivation**		M = 1.67	
**Aiming and trapping**		M = 0.65		**Attitude**		M = 1.70	
**Balance**		M = 0.76		**Attention/persistence**		M = 1.73	

**Table 2 ijerph-19-16835-t002:** ANOVA of the MABC-2 in relation to the age of the participants.

Dimensions	Age	Mean	SD	F	Sig.
**Manual skills**	4 years	0.92	0.205	1.688	0.191
	5 years	0.93	0.211		
	6 years	1.00	0.000		
**Aiming and trapping**	4 years	0.58	0.318	2.799	0.066
	5 years	0.60	0.409		
	6 years	0.77	0.311		
**Balance**	4 years	0.60	0.264	19.480	0.000
	5 years	0.73	0.285		
	6 years	0.96	0.100		

**Table 3 ijerph-19-16835-t003:** ANOVA for learning and gender.

Dimensions	Gender	Mean	SD	F	Sig.
**Motivation for competence**	Female	1.76	0.329	5.621	0.020
	Male	1.58	0.415		
**Attitude towards learning**	Female	1.75	0.262	1.995	0.161
	Male	1.66	0.315		
**Attention/persistence**	Female	1.87	0.335	10.045	0.002
	Male	1.59	0.506		

**Table 4 ijerph-19-16835-t004:** ANOVA for learning and age of participants.

Dimensions	Age	Mean	SD	F	Sig.
**Motivation for competence**	4 years	1.74	0.229	7.623	0.001
	5 years	1.47	0.516		
	6 years	1.80	0.244		
**Attitude towards learning**	4 years	1.79	0.217	12.721	0.000
	5 years	1.52	0.326		
	6 years	1.82	0.218		
**Attention/persistence**	4 years	1.80	0.386	7.267	0.001
	5 years	1.51	0.542		
	6 years	1.90	0.292		

**Table 5 ijerph-19-16835-t005:** Pearson’s correlation between learning and MABC-2.

	MS	AC	B	MOT	ATL	AP
**MS**	1					
**AC**	0.143	1				
**B**	0.347 **	0.435 **	1			
**MOT**	0.267 **	0.252 *	0.130	1		
**ATL**	0.437 **	0.322 **	0.336 **	0.532 **	1	
**AP**	0.377 **	0.309 **	0.290 **	0.801 **	0.731 **	1

Note. * Correlation is significant at the 0.05 level (bilateral); ** Correlation is significant at the 0.01 level (bilateral); MS, manual skills; AC, aiming and catching; B, balance; MOT, motivation for competence; ATL, attitude towards learning; AP, attention/persistence.

## Data Availability

The data used to support the findings of the current study are available from the corresponding author upon request.

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
