# Peer review of "Relationship between Learning and Psychomotor Skills in Early Childhood Education"

_ijerph, 2022, doi:10.3390/ijerph192416835_

Round 1

Reviewer 1 Report

Dear authors,

Thank you for submitting your manuscript to be reviewed. Although a descriptive study involving preschoolers’ motor skills and learning behaviors is relevant to the early childhood literature, this manuscript needs to be revised before being accepted for publication.

The introduction lacks focus and depth. The sentences are not connected, resulting in superficial information. The lack of flow causes confusion for the reader. My suggestion is to focus on the preschool’s contribution to the skills and behaviors addressed in the study. The text needs to provide a literature review of studies that assessed similar data. This study needs a framework to make the case that the data collected and analyzed are important to children’s development.

In addition, it would be important to have your manuscript proofread to assure that a more typical vocabulary is used. For example, the word ‘infant’ usually refers to babies, while the population you are referring to in this study is typically called ‘preschoolers.’ The MABC-2 test uses a beanbag (instead of a sack), aiming and catching as stated, but throwing, not shooting to a target. 

The methodology needs explicit description of the context and participants. Describe where the children are from and what kinds of activities the engage. If they are from the same school, how the school program looks like in terms of movement activities or physical play, add the ratio between teacher and students, etc. This is important, because all these factors may influence children’s development of motor skills. Who was part of the research team? 

Although the data found in the study seem to be substantial, without a substantial introduction, a strong statement of the purpose, and a thorough description of the context and participants and procedure, it doesn’t seem to be a significant contribution to the literature. 

Reviewer 2 Report

Dear Authors,

It was a great pleasure to read your article. The topic is very actual and important. And – what the most important – study design and research realization are very satisfactory, althou the conclusions part must be improved, and I have some other suggestion.Please find it below.

The title is informative and accurately reflects the manuscript. The abstract is complete and stand alone. It adequately reflect the content of the manuscript.

The Introduction provide sufficient theoretical background for the study. The theoretical framework is properly matched to the research problems being carried out – but please include a clearly defined problem at the end of the introduction part. It is missing. The introduction is structured logically. Definitions of all caterogies are present. The study’s justification is stated clearly.

 Please add more information about child care and Schools in Granada.

Please add information what etics committee approved the research (now we have only number).

The research design answers the proposed research questions. The study’s methodology and the execution of the study are adequate. The important aspects of the methods are clearly described.

The results are clearly organized and presented. The analysis are adequate described. Tables are clear and easy to interpret.

The structure of the Discussion is very clear. The interpretations are appropriate, supported by the results, and discussed with relevant literature.

Please add limitations of the study – I think that the last paragraph of conclusions are limitation. My suggestion is to write it down clearly.

The Conclusions section must be improved. It is too short and inconspicuous. How can these results be used in practice?

 Also, beceuse of a lack of information about the specific of child care and Schools in Granada, there is no information on how to use your results in this specific working environment. I find is as extremly important to improve.

Kind regards

Reviewer 3 Report

This study aims to describe psychomotor and learning behaviours of young children and examine the relationship between these two constructs.  These are worthwhile pursuits, but the challenge of descriptive studies is to ensure the sample captures diversity.  Typically, large stratified random sampling is used for this purpose.  Without this, the descriptives may be of interest to a local context such as a school interested in gaining a better understanding of the students, but it is not of broader interest. 

The procedure for obtaining the sample is not clearly described. I assumed all students attended one school.  If this is the case, there will be clustering issues that impact the relationships found between psychomotor and learning behaviours.  I don’t think there is any way to resolve this problem and it does make the findings very difficult to interpret and impossible to generalise.

Some minor points.

It would help to modify the language for an international audience.  For most, the term infant refers to the birth-24 month period.  The combination of infant, school and preschool in the abstract was confusing, although I’m sure it all makes sense within the Spanish education system.

I don’t think the statement on line 34 is either true (i.e. humans pre-civilisation could stand and grasp) or necessary to include.

Round 2

Reviewer 3 Report

Thank you for your response to the original review.  Your comment "The present study forms a part of a nationwide research project. The preliminary results reflect a first overview of schoolchildren in the city of Granada. Of course, given this research, these data cannot be generalised. The aim of the study is to detail a sample characteristics in order to make a comparison at the national level later on" indicates that the results, while publishable, are not sufficiently robust or at the level of interest for reporting in a high quality international journal.  Results from the larger study may be of more interest to an international readership.

Best of luck with your national project.
